# Systemic Sclerosis Patients Experiencing Mindfulness-Based Stress Reduction Program: The Beneficial Effect on Their Psychological Status and Quality of Life

**DOI:** 10.3390/ijerph20032512

**Published:** 2023-01-31

**Authors:** Khadija El Aoufy, Arianna Pezzutto, Alessandra Pollina, Laura Rasero, Stefano Bambi, Silvia Bellando-Randone, Serena Guiducci, Susanna Maddali-Bongi, Marco Matucci Cerinic

**Affiliations:** 1Department of Experimental and Clinical Medicine, University of Florence, Viale largo Brambilla 3, 50134 Florence, Italy; 2Department of Geriatric Medicine, Division of Rheumatology AOUC, 50134 Florence, Italy; 3School of Psychology, University of Padua, 35131 Padua, Italy; 4Center for Mindfulness Certified MBSR, University of Massachusetts, Worcester, MA 01605, USA; 5Department of Health Science, University of Florence, 50134 Florence, Italy; 6Unit of Immunology, Rheumatology, Allergy and Rare Diseases (UnIRAR), IRCCS San Raffaele Hospital, 20132 Milan, Italy

**Keywords:** Systemic Sclerosis, scleroderma, rheumatology, Mindfulness-Based Stress Reduction (MBSR), mindfulness, health-related quality of life

## Abstract

Psychological concerns in Systemic Sclerosis (SSc) patients represent an important issue and should be addressed through non-pharmacological treatments. Thus, the aim of the present study was to assess the effects of the Mindfulness-Based Stress Reduction (MBSR) program on psychological variables and the perspectives and experiences of patients with an SSc diagnosis. Notably, 32 SSc patients were enrolled and assigned to either the intervention (MBSR) group or the waitlist group. Inclusion criteria were (i) age ≥ 18 years, SSc diagnosis according to EULAR/ACR diagnostic criteria and informed consent. Exclusion criteria were previous participation in any Mind-Body Therapy or psychiatric diagnosis. Quantitative and qualitative outcomes were investigated through clinometric questionnaires and individual interviews. MBSR did not significantly impact outcomes such as physical functionality, anxiety, hopelessness, depression, physical health status, perceived stress, mindfulness and mental health status. For the anger evaluation, statistically significant differences are found for both controlling and expressing anger, indicating that the MBSR program had a favorable impact. As for qualitative results, more awareness of daily activities, stress reduction in terms of recognizing the causes and implementing self-strategies to prevent them, adherence to therapy, and recognition of the effect of medication on their bodies were reported. In conclusion, it is important to highlight the absence of negative or side effects of the MBSR program and the positive impact on patients’ experience and perspective; thus, we suggest this approach should be taken into account for SSc patients.

## 1. Introduction

Systemic Sclerosis (SSc) is a heterogeneous connective tissue, autoimmune and chronic disease with a variable and unpredictable course [1]. Increased deposition of extracellular matrix proteins leads to vascular lesions, immunological alterations and diffuse fibrosis of the skin and internal organs, causing the major symptoms of SSc [1,2,3]. The etiology of SSc is unknown and etiopathogenesis is not clear yet; however, the actual target of rheumatologists is, today, a very early diagnosis (VEDOSS) [4].

Patients with rheumatic conditions, in particular with SSc, report high levels of emotional distress compared to the general population due to the disease burden and the therapy burden; the level of emotional distress is different between diagnostic groups and individual patients and can vary over time [5]. Furthermore, negative emotions are common in patients with chronic illness and they can influence an individual’s functional status, symptom perceptions, and health-related quality of life (HRQoL) [6].

Another pivotal aspect is represented by patients’ perceptions according to their disease condition. Indeed, it is more associated with patients’ beliefs and emotional responses than with disease severity [7]. Thus, SSc patients can experience high levels of emotional distress, including depression, anxiety, fear about disease progression and the future, and body image concerns [8,9,10,11]. 

Unsurprisingly, psychosocial factors in rheumatic conditions are determinants of fatigue and may represent predictors of distress and depression more than clinical parameters [8]. Additionally, SSc patients frequently experience fears about disease progression, especially concerns about the risk of becoming disabled or fully dependent on others [11,12,13]. 

Lastly, physical disabilities associated with SSc have an important psychological impact, especially face and hands changes; hence, body image distress and concerns about appearance are common [11,14,15]. Thus, for the above-mentioned reasons, psychological concerns in SSc patients represent an important issue and should be addressed through non-pharmacological treatments [16,17]. The levels of HRQoL of SSc patients are significantly impaired when compared to healthy people [18,19] and are sometimes worse than other chronic conditions [20]. Thus, behavioral interventions designed to reduce emotional responses and negative appraisal of illness, such as Mind-Body interventions and Mindfulness meditation, could represent an effective and innovative approach for SSc patients. Mindfulness meditation has shown effectiveness in alleviating emotional distress and improving well-being in many populations [21,22,23]. In particular, Mindfulness-Based Stress Reduction (MBSR) was developed for patients with chronic pain and stress-related conditions [24,25], while today it is widely diffused and applied to many other conditions with successful results. There are a number of studies investigating MBSR’s effect on Fibromyalgia and Rheumatoid Arthritis but no study has focused on SSc patients [26,27,28]. Thus, the aim of the present study was twofold: (i) to assess the effects of the MBSR program on psychological variables, in particular anger, anxiety, depression, hopelessness and mindfulness, mental and physical health, and physical function, and (ii) to assess patients’ perspective and experience. 

## 2. Materials and Methods

### 2.1. Universe of Participants

We performed a prospective interventional study from November 2016 to February 2017 enrolling SSc patients from the Division of Rheumatology, Department of Experimental and Clinical Medicine, University of Florence.

Before participating in the study, all the patients signed an informed consent form. All the procedures were in accordance with the ethical standards of the 1964 Helsinki declaration. The study protocol was approved by the Institutional Review Board of the Regional Ethics Committee of the Region of Tuscany (protocol number: CEAVC SPE 16.310), Italy. The recruitment involved (i) Distribution of posters in the Day Hospital and in the Outpatient Clinic; (ii) Telephone calls; and (iii) One-day presentation of the MBSR program, organized in the Hospital. 

Inclusion criteria were (a)To have a physician’s diagnosis of SSc, according to criteria established by the American College of Rheumatology and European Alliance of Associations for Rheumatology [29]; (b) To be ≥18 years old; and (c) To sign the informed consent to participate.

Exclusion criteria were (I) To have previously participated in any Mind-Body Therapy, including MBSR programs; (II) To have a diagnosis of psychiatric disorders, according to criteria of the Diagnostic and Statistical Manual of Mental Disorders, Fifth Edition (DSM-V) [30].

Both groups were assessed before (T0, baseline) and after the intervention (T1, 8 weeks). After the intervention, we also investigated the perspective and experience of participants in the MBSR group using qualitative individual interviews led by a psychologist.

### 2.2. Mindfulness-Based Stress Reduction Program

The MBSR program consists of (i) 8 weekly meetings, typically lasting from 2.5 to 3 h, and (ii) “The silent day”, one full-day session taking place between the 6th and 7th meeting (see Figure 1). Participants practiced training in mindfulness, meditation, and gentle yoga and discussed the application of what they learned during the practice of mindfulness both in and out of the classroom (also used in the class active-listening techniques, in couples or small groups). Meditation was proposed in 4 different formats: (i) body scanning, an exercise to cultivate and strengthen attention by systematically and sequentially focusing on the sensations and feelings all along the various parts of the body, (ii) sitting meditation, in which the breath is used as an anchor of attention, (iii) contemplative walking and (iv) gentle Hatha Yoga postures, tailored to accommodate patients with SSc. Since the beginning of the 8-week training program, mindfulness activities were practiced both in class and as homework: participants were asked to practice 45 min of formal mindfulness exercises at home, every day during the 8-week period, helped by an audio CD. The application of mindful awareness to daily activities (called “informal mindfulness practices”) was also encouraged constantly. SSc patients were asked, during the 8-week period, to participate actively and also document their daily practice in a journal. Two certified MBSR teachers, trained at the “Centre for Mindfulness” at the University of Massachusetts, taught the class; they used standard course materials [31]. 

### 2.3. Questionnaires Application and Outcome Measures

Anger using the State-Trait Anger Expression Inventory-2 (STAXI-2). The STAXI-2 is a self-reported questionnaire with 57 items, on a 4-point Likert scale, to assess the following dimensions of anger: (i) state anger, (ii) trait anger, (iii) anger expression-in, (iv) anger expression-out, (v) anger control-in and (vi) anger control-out [32].

Anxiety using the State-Trait Anxiety Inventory (STAI-Y). The STAI-Y is a self-reported questionnaire with 40 items, on a 4-point Likert scale, to measure both state and trait anxiety. In STAI-Y1 (state anxiety-items 1–20), the intensity of feelings “in this moment” is assessed, while in STAI-Y2 (trait anxiety-items 21–40) the focus is on the frequency of feelings “in general [33].

Depression using the Center for Epidemiologic Studies Depression Scale (CESD). The CESD is a self-reported questionnaire, to assess depressive symptoms over the past week with 20 questions. The scores range from 0 to 60 and higher scores indicate higher levels of depression, with a cut-off for possible depression as ≥16 [34]. The questionnaire has been shown to be a reliable and valid measure of depressive symptoms in patients with SSc [35].

Hopelessness using the Beck Hopelessness Scale (BHS). The BHS is a self-reported questionnaire with 20 items designed to measure the construct of hopelessness in a true-or-false response format. It combines 11 negatively worded items with nine positively worded items. The scores range from 0 to 20 [36].

Mindfulness using the Cognitive and Affective Mindfulness Scale-Revised (CAMS-R). The CAMS-R is a self-reported questionnaire, on a 4-point Likert scale, to assess dispositional mindfulness. It is a short measure (only 12 items) and describes an attitude or approach toward the experience of one’s emotions or thoughts in 4 areas: focusing attention, being oriented to the present moment, being aware of an experience and having an attitude of acceptance or nonjudgment toward an experience. The scores range from 12 to 48: higher scores indicate a higher ability to be “mindful” [37].

Health-related quality of life (HRQoL) using the Medical Outcomes Study Short Form Healthy Survey (SF-36). The SF-36 is a self-reported questionnaire to assess the state of health with 36 questions exploring 4 physical health domains (physical activity, physical role, bodily pain, general health) and 4 mental health domains (vitality, social activities, role emotional, mental health). The SF-36 scales can be summarized into 2 overall indexes of physical health and mental health (SMI = summary mental index; SFI = summary physical index). The score ranges from 0 to 100, with a higher score representing a higher quality of life [38].

Physical functionality using the Health Assessment Questionnaire–Disability Index (HAQ-DI). The HAQ-DI is a self-reported questionnaire, ranging from 0 (no disability/best function) to 3 (maximal disability/worse function), with 20 divided into 8 domains: dressing and grooming, arising, eating, walking, personal hygiene, reach, grip strength and common daily activities [39]. 

Perceived stress using the Perceived Stress Scale (PSS). The PSS is a self-reported questionnaire with 10 items to assess stress during the last month (0 as “never” and 4 as “very often”). Scores ranging from 0 to 13 would be considered low stress, 14–26 would be considered moderate stress and 27–40 would be considered high perceived stress [40].

### 2.4. Statistical Analysis

Descriptive analysis of the two groups is presented as mean and standard deviation, *n* and percentages, median and interquartile range. Normality of distribution of all variables was assessed via the Shapiro–Wilk test. Student’s *t*-test was used and data were reported as mean and standard deviation to assess differences between groups at baseline and at T1 for parametric variables. While data were reported as median and interquartile range and the Mann-Whitney test was used to assess differences between groups at baseline and at T1 for non-parametric variables. Data were analyzed using Stata 13 (Statacorp LLC, College Station, TX, USA).

## 3. Results

Notably, 140 consecutive patients with SSc (129 women) from the outpatient clinic (Florence) were invited to participate in the study, and 32 of them met the inclusion criteria and were enrolled. Patients were assigned to either the intervention (MBSR) group (*n* = 16; all women) or the waitlist (control) group (*n* = 16; 15 women). The intervention group engaged in the MBSR program for 8 weeks, starting on March 2017. The quantitative results of our study are reported in Table 1 and Table 2, the first one describes the characteristics of the participants at baseline. We did not observe significant differences (all, *p*-value > 0.05) between the experimental and the control group at baseline. Then, the second part of the table shows the descriptive characteristics of the clinometric variables of the study participants in the MBSR program or control group, and no significant differences (*p*-value > 0.05) between groups are reported at baseline. Participants of the MBSR group were asked to report adherence to home exercises, and the mean adherence was 72% (range 39–100%), while class attendance was 93% (range 67–100%). While the second table shows the clinometric variables of the study participants in the MBSR program and the differences between groups after the intervention (T1). Physical functionality, anxiety, hopelessness, depression, physical health status, and perceived stress did not show a significant difference between groups (*p*-value > 0.05). Mindfulness and mental health status did not show a statistically significant difference after two months either. As for the anger evaluation, statistically significant differences, meaning the training with the MBSR program had a positive effect, are reported for anger control outside (*p*-value = 0.045), anger control inside (*p*-value = 0.009) and the anger expression index (*p*-value = 0.014). 

As for qualitative results, patients’ perspectives and experiences were investigated through qualitative individual interviews led by a psychologist, data were collected and the following key concepts were extrapolated:

Physical and emotional nonjudgmental awareness: “*I accept them as they are, trying not to judge myself or others, just listen to my body* (woman, 49 years)”. “*It is not easy, after a whole hectic life and a continuous struggle against time, to reduce the pace and think of myself in the present, to welcome the messages of my body and my mind, to look inside and around me with awareness, without mulling over the past, but I have already achieved some small results* (woman, 63 years)”. “*I realized how long I hurt myself by not listening to myself* (woman, 52 years)”. 

Reduced stress inputs: “*My back, neck and shoulders are less stiff and tight and the pain has lessened* (woman, 63 years)”.

Reduced/prevented panic attacks or just fear of the future: “*The fear of the future has subsided and the panic attacks have almost disappeared* (woman, 63 years)”. “*Fear destroys reason, tears the nervous system, immobilizes: those who live alone are more prey to it…The “weapons” I received in this Path, I will not keep them in a drawer, because they are now part of my daily living, they are irreplaceable: a gift that, with my years, I no longer thought I could receive* (woman, 81 years)”.

Taking medications with awareness: “*Being more aware of my actions, especially routine ones (like taking medication), helps me remembering what I’ve done and what I still need to do* (woman 60 years)”. Blesative resultsable 1 and 2 should be taken into account for SSc patients especially when theipsychological.

## 4. Discussion

The primary aim of the present controlled trial was to assess the effects of the MBSR program on psychological health (particularly, anger, anxiety, depression, hopelessness, mindfulness and mental health), physical health, physical function and the patients’ perspectives and experiences. The results of the study showed that the MBSR program has a positive and significant effect only on anger control and expression, as assessed by the STAXY scale, was reported. Additionally, qualitative data show that the participants experienced benefits in stress reduction, awareness of daily activities and treatment adherence after the 8-week period. 

To the best of our knowledge, this is the first study testing the effects of an MBSR program in SSc patients. We found that, generally, the MBSR program on our SSc patients did not have a significant effect on quantitative outcomes. However, previous evidences reported improvements in other RMDs. For instance, in fibromyalgia (FMS), an MBSR program improved the patients’ quality of life and health status [41,42,43]. Additionally, a positive patient acceptability of this type of program is reported in the literature. Additionally, a previous RCT in Rheumatoid Arthritis demonstrated that a standardized MBSR intervention resulted in reducing disease activity, morning stiffness and pain score [44]. In this line, we found that the MBSR program improved anger control and expression in our SSc patients. Accordingly, our previous pilot study on Sjogren Syndrome (SS) patients showed a significant improvement after the MBSR program for mental health, perceived stress and mood score which includes also anger [45]. However, given that the present study is the first evidence of the positive effects of an MBSR program on anger in SSc patients, further research is warranted.

To sum up, a recent systematic review of the literature showed that the effects of the MBSR program on fibromyalgia patients are still uncertain due to individual study limitations, inconsistent results and imprecision. Thus, suggesting further studies are necessary to address the size effect of MBSR [42]. In our SSc patients, the MBSR program did not improve any physical health outcome. Although a systematic review concluded that MBSR is effective in improving the physical health outcomes of asthma, chronic pain, tinnitus, fibromyalgia and somatization disorders, these conditions are not comparable to SSc. However, there are inconsistencies in the literature. For instance, Grossman et al. conducted a three-armed RCT in fibromyalgia and concluded that an MBSR program did not improve cardiac activity, respiratory activity and physical activity [46]. Additionally, first phase II clinical studies, e.g., on cognitive behavioral therapy and mind-body therapies (e.g., mindfulness-based stress reduction), show some benefits in stressful human diseases [47,48]. However, it is important to take many co-variables into consideration, such as disease duration, individual-related factors, medication and adverse events as well as comorbidities when dealing with psychosocial training. As for the impact of the MBSR program on the psychological and mental sphere, a recent study by Malakoutikhah et al. showed that an individual’s level of mindfulness can predict her/his level of general health, anxiety and anger [49]. SSc patients have to deal with a great impact not only on the physical disability but also on the psychological and mental aspects. Indeed, these are important aspects to take into account and assess regularly when taking care of patients with chronic and disabling diseases such as SSc. In our study, the MBSR program did not improve outcomes such as anxiety, depression, hopelessness, mindfulness, mental and physical health, and also physical function, which is in line with other studies in RA, FMS, SS and also SLE patients. However, different studies have found that mindfulness-based interventions are effective in improving physical and psychological symptoms, as well as general health, anxiety, and anger [50,51].

The secondary aim was to assess patients’ perspectives and experiences after participating in the MBSR program. In general, more awareness in daily activities was reported, that is, to “welcome both pleasant and unpleasant emotions and/or physical sensations”. Stress reduction was reported in terms of recognizing the causes and implementing self-strategies to prevent them; the awareness in daily activities was also linked to “taking medications”, resulting in patients’ adherence to therapy and recognition of the effect of medication on their bodies. In accordance with our results, other researchers suggested that activities such as body scans may help participants notice the pain they are experiencing without trying to change it. This attitude toward the pain, named detached awareness, helps the patients to shift their attention from the pain to the body itself, which can ultimately support pain relief and stress reduction [52]. A previous qualitative study in 6 SLE patients (another connective tissue disease) showed that an MBSR intervention improved patients’ ability to differentiate between themselves and the disease, their ability to accept, rather than actively fight, the fact that one must live with the disease and decreased behavioral avoidance [53]. Although, despite these promising results, no quantitative data were collected, and the sample size was too small to draw any conclusions. Indeed, they observed beneficial effects for their SLE patients, concluding that mindfulness practice might be a promising treatment option, especially because of its emphasis on the acceptance of negative physical and emotional states. 

Some study limitations need to be underlined. Patients were not randomly assigned to the treatment or the control group. Additionally, non-significant results may be due to the low sample size of the study, indeed larger randomized controlled trials could be useful to investigate the effectiveness of the MBSR program on physical and mental health outcomes of SSc patients. In summary, given the different pathophysiology, current evidence on RA, FMS or SS cannot be extended to SSc. Thus, further trials with robust study designs, preferably, randomized and larger sample sizes, are clearly needed. 

## 5. Conclusions

The experience of our patients regarding the MBRS program showed interesting aspects to take into account. Firstly, it is important to highlight the absence of negative or side effects; thus, this program appears to be safe. Secondly, the positive impact on patients’ perspectives and thoughts, resulting in better strategies to cope with the stress and other negative aspects related to their disease, represents an important, yet preliminary, outcome. In conclusion, we suggest that SSc patients could benefit from this mind-body approach; therefore, healthcare providers should take it into account when taking care of SSc patients.

## Figures and Tables

**Figure 1 ijerph-20-02512-f001:**
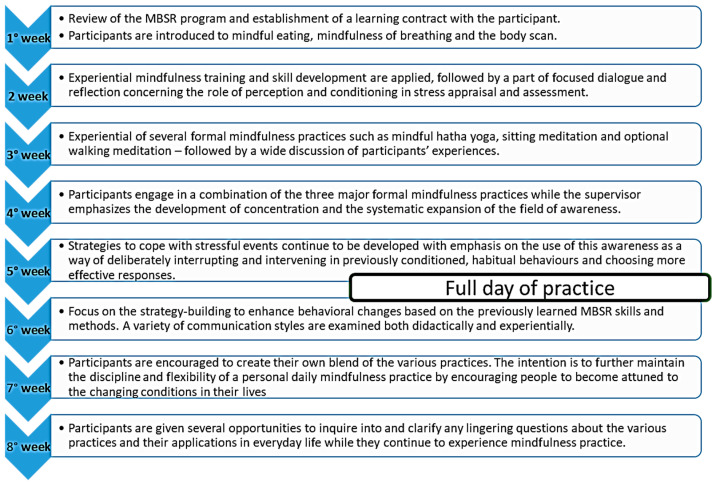
Activities performed during the sessions of the MBSR program.

**Table 1 ijerph-20-02512-t001:** Baseline descriptive characteristics on age, years of SSc diagnosis, education, occupation and clinometric variables of the study participants in the MBSR program (control and experimental groups) and differences between groups.

	MBSR Group (*n* = 16) % or M ± SD	Control Group (*n* = 16) % or M ± SD	*p*-Value
**Age (years)**	61.9 ± 12.9	54.5 ± 12.9	0.12
**Years of SSc diagnosis (months)**	80.3 ± 65.2	122.9 ± 86.4	0.15
**Years of education**	11.8 ± 3.3	12.06 ± 3.3	0.86
**Education NO**	18.75%	0%	0.25
**Education media**	18.75%	31.25%	
**Education college**	37.5%	56.25%	
**Education University**	25%	12.5%	
**No occupation**	25%	50%	0.13
**Yes occupation**	18.75%	31.25%	
**Retired**	56.25%	18.75%	
**Comorbidity YES**	56.25%	37.5%	0.47
**Comorbidity NO**	43.75%	62.5%	
**Physical Functionality (HAQ-DI)**	0.51 ± 0.59	0.21 ± 0.33	0.07
**State Anxiety (STAIY1)**	43.7 ± 12.8	46.1 ± 12.1	0.60
**Trait Anxiety (STAIY2)**	49.2 ± 12.9	45.09 ± 13.9	0.51
**Mindfulness (CAMS)**	72.6 ± 0.7	2.6 ± 0.7	0.96
**Hopelessness (BHS)**	6.4 ± 4.4	8.6 ± 6.0	0.26
**Depression (CESD)**	15.3 ± 8.2	19.7 ± 17.6	0.97
**Physical Health Status (SF36-SPI)**	39.7 ± 10.9	42.9 ± 10.6	0.27
**Mental Health status (SF3-SMI)**	44.3 ± 10.8	40.7 ± 13.1	0.41
**State Anger (STAXI_RS)**	18.4 ± 5.5	21.1 ± 12.3	0.92
**Trait Anger (STAXI_ RT)**	18.0 ± 4.5	19.5 ± 6.3	0.47
**Anger Expression outside (STAXI EROut)**	12.8 ± 2.6	13.1 ± 3.4	0.81

**Table 2 ijerph-20-02512-t002:** Clinometric variables of the study participants in the MBSR program and differences between groups after 2 months.

	MBSR Group (*n* = 16) M ± SD	Control Group (*n* = 16) M ± SD	*p*-Value
**Physical Functionality (HAQ-DI)**	0.46 ± 0.52	0.16 ± 0.40	0.08
**State Anxiety (STAIY1)**	37.1 ± 10.7	36.5 ± 23.1	0.92
**Trait Anxiety (STAIY2)**	41.5 ± 10.8	37.5 ± 23.8	0.56
**Mindfulness (CAMS)**	2.9 ± 0.5	2.2 ± 1.2	0.13
**Hopelessness (BHS)**	5.5 ± 3.1	8.1 ± 7.2	0.53
**Depression (CESD)**	12.7 ± 9.7	15.1 ± 15.6	0.96
**Physical Health Status (SF36-SPI)**	38.4 ± 10.4	42.8 ± 10.6	0.28
**Mental Health status (SF36-SMI)**	49.1 ± 9.8	40.8 ± 13.3	0.07
**State Anger (STAXI_RS)**	16.9 ± 3.7	17.6 ± 12.2	0.65
**Trait Anger (STAXI_ RT)**	17.1 ± 5.0	15.0 ± 9.9	0.45
**Anger Expression outside (STAXI EROut)**	13.1 ± 2.7	10.6 ± 6.0	0.50
**Anger Expression inside (STAXI ERin)**	16.9 ± 4.9	14.9 ± 9.0	0.45
**Anger Control outside (STAXI CROut)**	22.1 ± 3.7	16.1 ± 9.2	0.045
**Anger Control outside (STAXI CRin)**	25.3 ± 4.9	16.9 ± 9.9	0.009
**Anger Expression Index (STAXI ER Index)**	30.5 ± 11.3	40.3 ± 9.2	0.014
**Perceived Stress (PSS)**	18.6 ± 4.4	18.7 ± 4.9	0.97

## Data Availability

The data presented in this study are available on request from the corresponding author. The data are not publicly available due to privacy reasons.

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
