# Peer review of "Systemic Sclerosis Patients Experiencing Mindfulness-Based Stress Reduction Program: The Beneficial Effect on Their Psychological Status and Quality of Life"

_ijerph, 2023, doi:10.3390/ijerph20032512_

Round 1
Reviewer 1 Report
Dear authors,
The article presents interesting information. Several modifications are necessary:
In the abstract you do not include Objectives: (remove the word), methods: (remove the word), conclusions :..You simply state. In conclusion...
Line 28- rephrase
Line 82- par-ti-ci-pa-ting
Line 98- ma-jo-ri-ty
Line 120-helped by
Where is Figure 1?
The sample test is too small..You had only 32 pacients
Line 188-while class attendance was...
The conclusions section is too short. Please add more relevant information from your article

Author Response
In the abstract you do not include Objectives: (remove the word), methods: (remove the word), conclusions :..You simply state. In conclusion...
ANSWER: Thank you for your suggestion, we corrected the abstract as requested. (Abstract section)
Line 28- rephrase
ANSWER: For the anger evaluation, statistically significant differences are found for both controlling and expressing anger, indicating that the MBSR program had a favourable impact. Line 28
Line 82- par-ti-ci-pa-ting / Line 98- ma-jo-ri-ty /Line 120-helped by
ANSWER: we have corrected the typing errors above mentioned.
Where is Figure 1?
ANSWER: We apologize for the inconvenient. Figure 1 was a simple flow chart of patients’ enrolment, yet we decided to remove it as it did not add any information to the manuscript. However, we’ve added another figure one for the MBSR program activities.
The sample test is too small. You had only 32 pacients
ANSWER: We agree with the reviewer that the sample size is not large, however this is a pilot study and there are no other published data on the same topic. Thus, we believe that despite the small sample size our article reports new and interesting results.
Line 188-while class attendance was...
ANSWER: Corrected the sentence.
The conclusions section is too short. Please add more relevant information from your article
ANSWER: Thank you for your suggestion, the section conclusion is now longer, with few key concepts.
Reviewer 2 Report
This paper described a trial that assessed the effects pf the MBSR program on psychological health, physical function and the patients’ perspective and experience. 32 patients were included in this trial and were divided to either MBSR group or the control group. The MBSR program consist of 8 weekly meetings, results showed that participants experienced significant benefits on anger control and expression. This is the first paper investigating MBSR’s effect on SSc patients. This study provides some insights on this field.
I have some questions regarding the data collection: How many time points are you collecting assessments? Did you collect assessments like 3 months after the MBSR program?
Lines 73-74: please list the references that study the MBSR’s effect on Fibromyalgia and Rheumatoid Arthritis.
Line 81: Please clarify the definition of lcSSc and dcSSc, and the reason why most of the patients are women (ideally men patients and women patients are 50% each).
Line 101: There is one male patient in control group, I wonder if this one patient will affect the outcome, male patient is known to react different to stress and anger due to the hormone difference. I suggest rule out this male patient result and see the outcome.
Line 188 : while the class attendance was 93%.
Line 192: also and either please choose one.
Please clarify what is difference and relationship of State Anger/Trait Anger and Anger Expression/Control.
Author Response
Reviewer 2
This paper described a trial that assessed the effects pf the MBSR program on psychological health, physical function and the patients’ perspective and experience. 32 patients were included in this trial and were divided to either MBSR group or the control group. The MBSR program consist of 8 weekly meetings, results showed that participants experienced significant benefits on anger control and expression. This is the first paper investigating MBSR’s effect on SSc patients. This study provides some insights on this field.
ANSWER: Thank you for your accurate revision and your comments, which gave us the opportunity to improve our manuscript.
I have some questions regarding the data collection: How many time points are you collecting assessments? Did you collect assessments like 3 months after the MBSR program?
ANSWER: Many thanks for your meaningful suggestion. We did not provide any follow-up visits after T1, because this pilot study was designed also for reasons of feasibility in terms of available resources. However, when we replicate this research and this experimental program is definitively set up, a 3 months follow up will be performed for patients included in the program
Lines 73-74: please list the references that study the MBSR’s effect on Fibromyalgia and Rheumatoid Arthritis.
ANSWER: We added some references regarding mindfulness based interventions on RA and FMS patients. References 26-27-28
Line 81: Please clarify the definition of lcSSc and dcSSc, and the reason why most of the patients are women (ideally men patients and women patients are 50% each).
ANSWER: We have decided to leave the term “Systemic Sclerosis” –SSc only, as specifying limited or diffused has no relevance for this study. Thanks for pointing that out. As for the presence of female or male patients, it is widely documented that the ratio F:M that is 9/10:1.
Line 101: There is one male patient in control group, I wonder if this one patient will affect the outcome, male patient is known to react different to stress and anger due to the hormone difference. I suggest rule out this male patient result and see the outcome
ANSWER: Thanks for your comment; the sample was a convenience sample and we did not place men or women in the treatment or in the control group. Also, as the disease is prevalently present in the female population, the numbers of our study represents real life. It is, however, interesting in future studies to check the different reaction of men and women to this types of mind-body therapies
Line 188: while the class attendance was 93%.
ANSWER: Thanks, we corrected the sentence.
Line 192: also and either please choose one.
ANSWER: Thanks, we corrected the sentence.
Please clarify what is difference and relationship of State Anger/Trait Anger and Anger Expression/Control.
ANSWER: State anger as the experience of negative feelings similar to being annoyed or irritated, or to a greater extent, filled with rage, while he describes trait anger as how frequently state anger is experienced. Then, expressing angry feelings in aggressive verbal or motor behavior directed toward other people or objects in the environment (angerout) or suppressing those feelings and holding them in (anger-in) are the two modes of anger expression. The scale was designed to give as an overview of how the respondent deals with all the aspects related to anger.
Reviewer 3 Report
The present work focuses on a highly relevant topic, such as the challenge/problem of addressing the psychological concerns of systemic sclerosis patients from pharmacological treatments. Specifically through the evaluation of the effects of the Mindfulness Based Stress Reduction (MBSR) program on psychological variables and the perspective and experience of patients diagnosed with SSc.
The aim of the work is clear and the methodology and analysis allows it to be achieved. However, there are aspects that must be improved.
In the introduction it would be important to highlight:
- What are the psychological ailments and effects of SS on patients. Reference
- The application of Mindfulness-Based in similar diseases or other cases from which effects could be extrapolated. Reference.
- The material and method section is somewhat disorganized. It is difficult to follow.
It should follow the logic of the funnel, from the most general to the particular. In this sense, it should begin by describing in broad strokes the type of study and include a visual diagram that structures the stages of the research (to facilitate comprehension and follow-up by the reader).
For example: Stage 1: Universe of participants, Stage 2 Mindfulness Based Stress Reduction program, Stage 3 Questionnaire application, Stage 4 Statistical analysis.
It is not clear whether the application of questionnaires is done during or after the program.
In the 2.1. Mindfulness Based Stress Reduction program a table or infographic is needed to allow a quick and accurate understanding of the total time of the program, the weekly dedication, hours, and the different types of meditation performed.
The conclusion is extremely synthetic and lacks to extract the most relevant aspects of the discussion.
In the textual quotations. It is important to homogenize the style. Some quotations are in italics and others are not.
When including references in the text, they should be in brackets [1], [2], [3] and not in parentheses. (1),(2),(3).
Author Response
Reviewer 3
The present work focuses on a highly relevant topic, such as the challenge/problem of addressing the psychological concerns of systemic sclerosis patients from pharmacological treatments. Specifically, through the evaluation of the effects of the Mindfulness Based Stress Reduction (MBSR) program on psychological variables and the perspective and experience of patients diagnosed with SSc.
The aim of the work is clear and the methodology and analysis allows it to be achieved. However, there are aspects that must be improved.
ANSWER: We answered all the comments and we thank the reviewer for the comments and revision, as it gave us the opportunity to improve our manuscript.
In the introduction it would be important to highlight:
- What are the psychological ailments and effects of SS on patients. Reference
ANSWER: References 8 to 15
- The application of Mindfulness-Based in similar diseases or other cases from which effects could be extrapolated. Reference.
ANSWER: We added the references, as suggested.
- The material and method section is somewhat disorganized. It is difficult to follow. It should follow the logic of the funnel, from the most general to the particular. In this sense, it should begin by describing in broad strokes the type of study and include a visual diagram that structures the stages of the research (to facilitate comprehension and follow-up by the reader). For example: Stage 1: Universe of participants, Stage 2 Mindfulness Based Stress Reduction program, Stage 3 Questionnaire application, Stage 4 Statistical analysis.
ANSWER: Thanks for your suggestion, we have modified the material and method section and we hope it is clearer now. Also, for the MBSR program we added a figure to better explain the protocol that SSc patients underwent.
It is not clear whether the application of questionnaires is done during or after the program.
ANSWER: The administration of the selected questionnaires was done before the first session and after the last session of the MBSR program, referred as baseline and T1 (8weeks), also we added this detail in figure 1.
In the 2.1. Mindfulness Based Stress Reduction program a table or infographic is needed to allow a quick and accurate understanding of the total time of the program, the weekly dedication, hours, and the different types of meditation performed.
ANSWER: We added a figure in which we put all the content details of the activities included in the MBSR program. (see figure 1)
The conclusion is extremely synthetic and lacks to extract the most relevant aspects of the discussion.
ANSWER: Thanks for your suggestions, indeed the conclusion were extremely short. We have revised the conclusion as suggested.
In the textual quotations. It is important to homogenize the style. Some quotations are in italics and others are not.
ANSWER: Typing error, we corrected and made it homogenous as all quotation are now in italics.
When including references in the text, they should be in brackets [1], [2], [3] and not in parentheses. (1),(2),(3).
ANSWER: Thanks for pointing that out, we have now substituted all the parentheses with brackets for the references.
Round 2
Reviewer 3 Report
The authors made substantial improvements based on suggestions.